# Endothelial Function: A Short Guide for the Interventional Cardiologist

**DOI:** 10.3390/ijms19123838

**Published:** 2018-12-02

**Authors:** Tommaso Gori

**Affiliations:** Kardiologie I, Zentrum für Kardiologie der Universitätsmedizin Mainz and DZHK Standort Rhein-Main, Langenbeckstr 1, 55131 Mainz, Germany; tommaso.gori@unimedizin-mainz.de; Tel.: +49-6131-172829

**Keywords:** endothelial function, shear stress, coronary stenting

## Abstract

An impaired function of the coronary endothelium is an important determinant of all stages of atherosclerosis, from initiation, to mediation of functional phenomena—such as spasm and plaque erosion, to atherothrombotic complications. Endothelial function is modified by therapies, including stent implantation. Finally, endothelial function changes over time, in response to physical stimuli and pharmocotherapies, and its assessment might provide information on how individual patients respond to specific therapies. In this review, we describe the role of the endothelium in the continuum of coronary atherosclerosis, from the perspective of the interventional cardiologist. In the first part, we review the current knowledge of the role of endothelial (dys)function on atherosclerotic plaque progression/instabilization and on the mechanisms of ischemia, in the absence of coronary artery stenosis. In the second part of this review, we describe the impact of coronary artery stenting on endothelial function, platelet aggregation, and inflammation.

## 1. Introduction

The coronary endothelium lines the internal lumen of the heart blood vessels, and is constituted by single cell layer that separates the blood from the vascular media. Tight, adherens and gap junctions maintain the integrity of this monolayer. Its multiple functions include anti-coagulant properties, fluid and nutrient trafficking, the regulation and repair of vascular homeostasis, and the control of vascular tone. A dysfunction in any of these properties is considered to be a prerequisite and a necessary initial step of coronary artery disease. “Endothelial function” is traditionally assessed by investigating only one of these functions (e.g., vasomotor control or release of adhesion molecules), an assumption that surely represents an oversimplification of the homeostasis of a complex organ. It is well-known that, virtually, all cardiovascular risk factors cause endothelial dysfunction by altering various molecular pathways, among which the induction of oxidative stress is considered to be a prevalent one [1,2,3]. An increased production of oxygen free radicals, from multiple sources (including the uncoupled mitochondrial respiration, NAD(P)H oxidases, and the uncoupled endothelial nitric oxide synthase), results in an impaired endothelial homeostasis, with a switch from an anticoagulant, vasodilator, anti-inflammatory phenotype to a pro-coagulant and pro-inflammatory vasoconstrictor, one mediated by an increased bioavailability of peroxynitrite, NO (nitric oxide) synthase uncoupling, prostacyclin formation inhibition, endothelin expression stimulation, and a reduced NO signaling, following the inhibition of soluble guanylate cyclase activity [4,5] (Figure 1). Superoxide anions from these multiple sources react with endothelial NO producing peroxynitrite (ONOO−). Beyond this direct scavenging effect, ONOO− interferes with the endothelial NO synthase activity, by oxidizing its cofactor tetrahydrobiopterin (BH4). The resulting BH3 radical competes with BH4 and prevents the transfer of electrons from NADPH to arginine, resulting in further superoxide production (“eNOS uncoupling”). A more detailed description of the mechanisms of endothelial dysfunction, and their implications for cardiac and vascular function, eludes the scope of the present review and has been recently published [6]. Folic acid has been proposed to reverse these phenomena [7], but the negative data on the clinical impact of antioxidant therapies appear to suggest that, a simple removal of reactive oxygen species, is not sufficient to reverse these processes and prevent their implications [5].

There are at least four pathways through which an impaired endothelial function may affect the genesis and progression of coronary artery disease. First, compatible with Libby’s and Maseri´s paradigm of atherosclerosis as an inflammatory disease, is the induction (or withdrawal of inhibition) of inflammatory reactions that lead to the formation of coronary plaques [8]. The second pathway is the dysregulation of vascular tone, causing spasm phenomena, in the absence of atherosclerotic plaques. The third pathway is thrombogenesis, and, finally, the fourth is the induction (or reduced inhibition) of cellular proliferation, after coronary stenting.

Since myocardial oxygen extraction is already virtually submaximal, in resting conditions, any additional demand must be compensated by an increased coronary flow. In healthy individuals, a normal coronary blood flow reserve is at least three times the resting blood flow. Such increased blood flow is achieved by dilation of resistance arteries and inhibition of spasm of epicardial vessels. Release of endothelial vasodilator substances, such as NO and prostacyclin, as well as the inhibition of endothelial vasoconstrictor substances (mainly endothelin-1), all triggered by shear stress in conduit arteries, and by ADP or changes in pH in the peripheral circulation, are the major mediators of these phenomena. Any of the above four mechanisms has the potential to reduce coronary flow reserve, in the presence or in the absence of coronary artery plaques. Collectively, these mechanisms contribute to explain the incomplete correlation between the angiographic evidence of coronary artery disease and the patients’ prognosis, and the frequent finding of myocardial ischemia in the absence of atherosclerotic lesions [9,10].

## 2. Endothelial Dysfunction as a Mechanism and Diagnostic/Prognostic Marker of Coronary Atherosclerosis

Atherosclerosis is a progressive, non-remitting disease whose prodromes are already visible, early in life, and is characterized by a long, subclinical, initial phase and a slow progression with multiple, to date unpredictable, phases of acceleration [11]. Reflecting its biological nature of systemic, organ-wide inflammation (rather than being a local phenomenon), classical observations demonstrate that an average of 2 (range 0–6) ruptured plaques can be found in each acute coronary syndrome patient, most commonly in different vessels [12]. Endothelial dysfunction, whether demonstrated by an impaired release of endothelial autacoids or an altered responsiveness to endothelium-dependent vasomotor stimuli, is present long before the anatomo-pathological or angiographic evidence of atherosclerosis appears. Coronary vasomotor dysfunction, in response to acetylcholine, precedes the development of coronary stenoses and is a predictor of prognosis, in patients with initial coronary artery disease [13]. Endothelial dysfunction has been reported in association with most known risk factors for atherosclerosis and, more importantly, rather than a marker of coronary artery disease (CAD) risk, it is considered to be a mechanism with an essential role in the genesis, progression, and instabilization of atherosclerosis by promoting coagulation, vasoconstriction, and impairing the vascular repair [4,14,15,16,17,18]. Even in the absence of clinically-relevant (flow-limiting) stenoses, endothelial dysfunction, itself, is indeed an acknowledged source of myocardial ischemia. In particular, over 20% of the patients referred for coronary angiogram due to chest pain, do not present significant obstructive coronary stenosis while suffering from typical angina with documented myocardial perfusion defects [19]. By inducing inappropriate vasoreactive phenomena, or increasing the propensity to spontaneous dissections and endothelial erosions, endothelial dysfunction is, therefore, a frequent cause of coronary pathology, independent of atherosclerosis, particularly in younger subjects or females. For instance, in a recent registry of 1379 consecutive patients with stable angina and unobstructed coronaries, Ong et al. reported that as many as 813 patients (59%) presented a pathological endothelial function test, of which 33% were at the level of the microcirculation and 26% at the epicardial level [20].

Endothelial dysfunction constitutes a first stage of “traditional” atherosclerosis, as it summarizes the effect of all cardiovascular risk factors. An abnormal endothelial vasomotor function correlates with the severity of coronary atherosclerosis (in coronaries more than in the cerebral circulation) and, using reclassification statistics, it improves the capacity of accepted risk scores to predict the presence and extent of CAD [21,22]. Additionally, the measurement of endothelial function in the peripheral circulation (e.g., at the level of the forearm) is an easily accessible surrogate of central endothelial function and it appears to provide information that has a clinical value comparable to that provided by the study of central (coronary) endothelial function—there is an association between the forearm and the coronary responses to the endothelial agonists [2] and the prognostic value of the peripheral versus the coronary endothelial responses, is similar. Further, in primary prevention, the endothelial function is also a long-term predictor of cardiovascular events, after adjusting for other risk factors [23], even though several studies question the additional benefit of an endothelial function assessment over validated clinical risk scores [24,25]. 

In sum, the assessment of endothelial function has the theoretical potential to serve as a clinical tool to re-classify the risk of the patients beyond the conventional risk factors. The real question is, however, not whether a population who shows evidence of endothelial dysfunction has an increased risk, but whether evidence of endothelial function has a practical clinical meaning, in a given patient [26]. Importantly, while this technique showed to be reliable in individuals at low-risk, it exhibited a poor accuracy in individuals at high-risk, in whom the structural abnormalities are already recognizable, with a poor long-term prognostic value in such patient subsets [14]. Almost three decades after the introduction of the endothelial function measurements, even without considering practical issues such as repeatability and standardization of the measurements, their utility in routine clinical settings remains, therefore, at least questionable. 

## 3. Endothelial Dysfunction after Coronary Artery Stenting

Coronary stenting is a mechanical therapy for CAD. Although necessary to restore the blood flow, stenting disregards the pathophysiology of CAD, and has profound implications on the vascular physiology. Balloon angioplasty and stenting produce acute endothelial injury, which invariably results in a dissection of the atheromatous plaque and stretching of the media and adventitia, and lysis of some of the cells, mainly vascular smooth muscle cells (VSMCs), or even intramural hematoma. 

These processes have not only mechanical implications, as they cause an increased local release of vasoconstrictive agents, such as serotonin and thromboxane, resulting in a vasospasm phenomenon [27]. Further, the endothelial denudation leads to exposure of tissue factor and the subendothelial extracellular layer, which triggers platelet deposition and attraction of leukocytes [28]. Platelet release of chemotactic factors and mitogens, such as the platelet-derived growth factor (PDGF), favors the proliferation, migration, and shift from a contractile to a synthetic phenotype [29]. On a longer term, the reduced inhibition of the VSMC proliferation and the increased deposition of the extracellular matrix proteins triggers intimal thickening, neointimal hyperplasia, and is one of the major risk factors for the development of stent restenosis. The mechanical stress also results in the release of VEGF and FGF, which favor, respectively, strut healing (see below) but also the proliferation of the smooth muscle cells. Percutaneous coronary intervention (PCI) also requires implantation of a prosthetic metallic foreign body, which interacts with both the arterial wall and the vessel lumen, invariably resulting in damage to the endothelium. Further, in the case of drug-eluting stents, the additional effect of the eluting polymer or the eluted drug must be accounted for. Finally, the stent struts cause an obstacle to blood flow, modifying blood rheology and fluid dynamics. The endothelium that covers stented segments shows typical areas of poor endothelialization, poorly-formed intercellular junctions, a reduced expression of anti-thrombotic molecules, and a decreased nitric oxide production [30].

### 3.1. Impact of Stenting on Fluid Dynamics and Shear Stress

Vascular cells, and endothelial cells, in particular, are exposed to the mechanical stimuli which modify their function. On the luminal side, endothelial cells sense shear stress; additionally, the wall strain due to the cyclical variations in vascular tone is also an important mechanical trigger for both endothelial and smooth muscle cells. The importance of these factors in the pathophysiology of the CAD, is observable from the preference that lesions show for the bifurcation segments, where blood flow is more turbulent and shear stress is lower [31,32].

In a similar way, the blood flow dynamics is modified by stent implantation in, at least, two ways. First, the mechanical stress of balloon inflation causes endothelial denudation and exposes the media to platelets and coagulation factors. Additionally, the stent struts protrude into the lumen, resulting in the areas of low and/or oscillating shear stress, which promote inflammation and vascular injury (Figure 2).

While the introduction of drug eluting stents has significantly limited the occurrence of restenosis and stent thrombosis, through a biological process (pharmacological inhibition of cell proliferation), it has become clear that the mechanical properties of the stent strut (including their dimensions, the distance between the struts and the distance between struts and the vessel wall) affect the stent thrombogenicity, even more than the presence of drug coatings [33].

Both in bare metal stents and drug eluting stents, an inverse correlation between the shear stress and neointimal hyperplasia has indeed been shown [34,35], with a particular effect at the level of the stent edges and in the cases of incomplete stent apposition or expansion. At a local level, thicker stent struts cause small regions of downstream flow reversal and a disturbed shear stress associated with platelet aggregation; at the top of the struts, areas of increased shear stress are associated with platelet activation; malapposed struts cause a mix of both phenomena [36]. 

The molecular mechanisms are only partially known—shear regulates a number of genes, mostly mediated by the pathway of the mechanosensitive transcription factors Kruppel-like factor-2 (KLF-2) and nuclear factor erythroid 2-related factor (Nrf2). In normal conditions, high, unidirectional shear stress activates these factors, which cooperate to induce the anti-inflammatory, anti-thrombotic, and anti-proliferative genes [37]. KLF-2 also regulates the vascular tone, via the endothelin-1 (ET-1) and endothelial nitric oxide synthase (eNOS) pathways, and it inhibits the NF-κB, thereby, reducing inflammation [38]. Nrf2 is involved in the induction of the antioxidant genes, including heme oxygenase-1, glutathione *S*-transferase, and ferritin. Further, it confers protection to the endothelial cells, by inhibiting the p38 MAPK pathway, which regulates apoptosis, proliferation, and inflammation [39].

In contrast, low and/or oscillatory shear stress, such as that found downstream to thick stent struts, activates the transcription factor NF-κB pathway, which controls multiple processes, including immunity, inflammation, cell survival, differentiation, and proliferation [40]. Further, it activates the c-Jun N-terminal kinase (JNK) and the p38 pathway, also promoting inflammation and apoptosis via the activation of the transcription factors of the activating protein-1 (AP-1) superfamily (including c-Jun and activating transcription factor-2 (ATF2)). An inhibition of JNK/p38 MAPK is achieved by high shear stress, via the induction of the MAPK phosphatase-1 (MKP-1). 

Low and/or oscillatory shear stress activates the transcription factor NF-κB [41], which controls immunity, inflammation, cell survival, differentiation, and proliferation, and it activates c-Jun N-terminal kinase (JNK) and the p38 pathway, promoting inflammation and apoptosis [42]. Finally, low and/or oscillatory shear stress inhibits the migration of cells, thus, preventing stent healing, and activates Akt, in the absence of AMPK, resulting in a sustained activation of p70S6K and subsequent endothelial cell proliferation [43]. 

### 3.2. Impact of the Stent on Endothelial Function

Older reports of impaired endothelial vasomotor responses after balloon dilatation, BMS implantation, and brachytherapy, led to the hypothesis that the mechanical injury associated with PCI could be responsible for the impaired vascular function [44,45,46] (Figure 3). Importantly, endothelial dysfunction, as assessed by the brachial flow-mediated dilation, has been reported to be a predictor of in-stent restenosis. This phenomenon became, however, more evident in studies which showed a more severe and persistent impairment in endothelium-dependent vasoreactivity in coronary arteries, after a drug eluting stent (DES) implantation [46,47,48,49,50,51,52]. In these studies, a DES of first-generation, provoked an endothelial dysfunction that was protracted for over six months after PCI. In contrast, a second-generation stent appeared to have a more benign impact. These observations were compatible with the lower incidence of stent thrombosis, despite a reduced antiplatelet therapy in the newer generation devices [53]. 

There are at least four different mechanisms that can explain the endothelial dysfunction induced by a DES.

#### 3.2.1. The Effect of Strain

Much less data are available concerning the impact of stent-induced strain on endothelial cells. Endothelial cells have multiple mechanoreceptors (including cell-adhesion sites, integrins, tyrosine kinase receptors, ion channels, and components of the lipid bilayer), which, upon activation, modify the activity of multiple signaling pathways (including the PKC, Rho, Rac, PI3K/Akt, and MAPKs), as well as transcription factors (including the AP-1 and NF-κB), and inflammatory genes (including ET-1, VCAM, and MMPs) [54]. Additionally, the excessive strain induces endothelial Ang II release and AT1R activation, resulting in vasoconstriction, inflammation, and elevated superoxide levels and simulating endothelial dysfunction from other sources.

#### 3.2.2. Biological Effect of the Drug 

The cytostatic drug eluted by the DES, is designed to reduce the risk of restenosis, by inhibiting neointimal proliferation. Since the same effect is also exerted on endothelial cells, however, delayed endothelialization has been reported in the DES, an observation that appears to be associated with increased incidence of stent thrombosis in the case series [55,56]. This impaired endothelial repair results in a reduced endothelial nitric oxide production and reduced endothelial inhibition of platelet aggregation. Further, sirolimus induces the release of vasoconstrictors and proaggregant autacoids, such as endothelin, and is associated with an increased endothelial production of reactive oxygen species by mitochondria [57,58]. Paclitaxel enhances the Rho-kinase expression and activity, stimulating inflammatory responses and microthrombus formation. Of note, the hydrophobic nature of sirolimus and paclitaxel allows them to penetrate the vessel wall and induce these deleterious effects, for prolonged periods of time [59].

#### 3.2.3. Hypersensitivity Reactions

Hypersensitivity reactions to the polymer of first-generation DES, with subsequent vessel enlargement, and severe localized hypersensitivity reactions predominantly consisting of T lymphocytes and eosinophils, has been repeatedly reported with first-generation DES and, particularly, with the Tacrolimus-eluting DES. The idea is that the cytotoxic drugs eluted by the first-generation stents, may result in an impaired strut healing [58], and that polymer residuals may be associated with the chronic inflammation/hypersensitivity reactions [60,61], which lead to extensive vasculitis of the intima, the media, and the adventitia. Histopathologically, these lesions consist mostly of lymphocytes and eosinophils, but mast cells and eosinophils may also participate. Aneurysmal coronary dilation is often present, resulting in stent malapposition and worsening of the shear stress modifications induced by the struts. The formation of thick fibrin bridges between the stent and the arterial wall, further impairs the fluid dynamic, leading to an impaired endothelialization, an adverse vessel remodeling, and neoatherosclerosis [62]. These modifications have also been attributed a causal role in the increased rates of very late stent thrombosis [63,64,65] with these generation of devices. The Kounis, or hypersensitivity-associated acute coronary syndrome, is a particular form of hypersensitivity characterized by the concomitant occurrence of acute coronary syndrome with allergic, hypersensitivity, anaphylactic, or anaphylactoid reactions. Arachidonic acid metabolites (such as, leukotrienes and thromboxane), proteolytic enzymes (such as, chymase and tryptase, histamine, cytokines, and chemokines), and inflammatory cells (such as, macrophages, T-lymphocytes, and mast cells) have all been involved, although, the molecular mechanisms and triggers are not yet known. Mast cells usually exist within the atherosclerotic lesion and release massive quantities of inflammation mediators, such as cytokines and chemokines, leading to platelet activation. Importantly, Hamilos et al. showed that endothelial function is less impaired by the modern-generation DES, particularly, those with a bioabsorbable polymer or the fluoropolymers of cobalt- and platinum-chromium DES, which have been shown to inhibit platelet and inflammatory cell adhesion, a phenomenon called fluoropassivation [66,67].

### 3.3. Impact of Stenting on Platelet Function

Endothelial cells, monocytes, and platelets are actively involved in the vascular responses to the trauma induced by coronary angioplasty and stenting. The mechanical injury induced by stenting causes platelet activation, degranulation, and the release of platelet activators, including ADP. Proinflammatory molecules released by platelets (including platelet factor 4, macrophage inflammatory protein 1, and RANTES) participate in these mechanisms [57,68,69,70]. The resulting thrombocyte activation causes further endothelial release of oxidative free radicals and inflammatory and mitogenic mediators, which stimulate the chemotaxis and transmigration of monocytes to the vessel intima [71,72]. Platelets expressing the CD40 ligand, further trigger the endothelial activation and the expression of adhesion molecules, chemokines, and the tissue factor, on the endothelial cells [73]. The reduced bioavailability of nitric oxide, which follows the scavenging by reactive oxygen species, further stimulates platelets, which are otherwise inhibited by NO. The cyclooxigenase-1 contained in the platelet cytoplasma, is also an additional source of free radicals and isoprostanes, a stimulus for further vascular production of reactive oxygen species from the vascular wall. Interestingly, this positive feed-forward mechanism leading to the endothelial dysfunction is interrupted by the administration of the platelet P2Y12 receptor inhibitors. In particular, ticagrelor might also have positive effects mediated by the stimulation of the adenosine signaling pathway, which results from the inhibition of the adenosine transporter [74]. Adenosine is a peripheral vasodilator which has ancillary properties, including the inhibition of the neutrophil trafficking and effector functions, the production of inflammatory mediators and granular release, and stimulation of the endothelial progenitor cells migration [75,76]. In a porcine model, ticagrelor reduced the neointimal formation and improved the endothelial function after a DES implantation, as compared to both clopidogrel and prasugrel. Additionally, it reduced the peri-strut chronic inflammatory cell infiltration [77]. As such, this class of molecules may have an important impact on endothelial function, and, therefore, on the incidence of stent thrombosis and in-stent restenosis (and, ultimately, on the patients´ outcome).

## 4. Endothelial Regrowth after Stenting

Most of the DES, currently available, release inhibitors of the mammalian target of rapamycin (mTOR), such as sirolimus or its derivatives (such as evero­limus, zotarolimus, or biolimus), for the ability of these drugs to inhibit the proliferation of smooth muscle cells and, consequently, to inhibit restenosis. The mTOR signaling pathway controls protein synthesis and it modulates cell division in response to mitogenic stimuli, through formation of the TORC1 complex, which includes mTOR, Raptor, and mLST8 proteins. In turn, TORC1 is responsible for the activation of downstream effectors like the p70 ribosomal protein (p70s6k1 and p70s6k2) and the eukaryotic translation initiation factor 4E-binding protein 1 (4E-BP1) [78]. On the other side, simultaneous inhibition of the TORC1 and TORC2 and their components (in particular p70s6k) might also have important negative consequences, as both of them are necessary mediators of endothelial regrowth, function, and survival.

Rapamycin inhibits cell motility by suppressing the mTOR-mediated S6K1 and 4E-BP1 pathways, by preventing the formation of TORC1, and by inhibiting the formation of the TORC2 complex, which consists of mTOR, the rapamycin-insensitive companion of the mTOR, SIN1, and mLST8. The TORC2 complex regulates the Akt activity. In contrast, TORC1 modulates the expression and activity of the hypoxia-inducible factor α (HIF-1α), which is one among the most potent transcriptional inducers of VEGF expression. Stent implantation leads to the upregulation of HIF-1α, which in turn leads to increased VEGF levels; both proteins are important mediators of vas­cular healing [79]. Other effectors that are downstream to TORC1, include the p70s6k (which stimulates endothelial cell proliferation), the p70s6k, and the 4E-BP1, which are involved in cell migration [80]. In turn, the TORC2 modulates the Akt-dependent pathway, which is responsible for the stimulation of endothelial proliferation, migration, and survival. Further, the phosphorylation and activation of eNOS, and the subsequent increased nitric oxide production, are mediated by TORC2 [81,82]. Thus, these data suggest a critical role for both the TORC1 and the TORCs in endothelial recovery and function.

The regeneration of the endothelial layer from the neighboring unin­jured segments, proximal and distal to the stented segment, as well as the migration and proliferation of the circulating progenitor cells, is necessary for the strut healing [83]. A delay in the process of the endothelial recovery is a known risk factor of late stent thrombosis [84] and is accelerated in the presence of incomplete strut apposition. As a consequence of the processes described above, this neoendothelium appears to be less mature, as shown by electron microscopy, and less functional than the original one, as shown by the evidence of impaired endothelium-dependent vasodilatation and increased permeability [85]. The adhesion of small platelet aggregates and monocyte attachment is associated with a reduced expression of PECAM-1, a phenomenon that is still present 28 days after the DES implantation and is associated with less-effective junctions between the endothelial cells [86]. By contrast, the antithrombotic cofactor thrombomodulin was less expressed on the neoendothelial surface, following an implantation of first-generation DES. Collectively, these data support the idea that the neoendothelium covering first-generation DES is less mature and lacks the antithrombotic features, as compared to that covering bare-metal stents. Further, the expression of VEGF, the growth factor responsible for endothelial-cell proliferation and permeability, appears to be increased, following implantation of sirolimus-eluting stents, paclitaxel-eluting stents, and zotarolimus-eluting stents, as compared to the everolimus-eluting stents and bare-metal stents. This persistence in the expression of VEGF, shows a transitional stage of immature re-endothelialization, following implantation of some, but not all, DES. Interestingly, similar differences have been shown in the incidence of stent thrombosis, and everolimus-eluting stents have shown the lowest rates of this potentially fatal complication [87]. In analogy, zotarolimus-eluting stents, which have also shown to be associated with a low incidence of stent thrombosis, also when a short dual antiplatelet therapy is administered [87], showed less-delayed healing and greater expression of eNOS, compared to the first-generation DES, in a rabbit model of atherosclerosis [88].

Shear stress also influences proliferation and migration of the endothelial cells after stenting. High, unidirectional shear stress increases the migration of cells and repair of the damaged endothelial layer, by promoting a remodeling of the actin cytoskeleton. The cytoskeleton controls cell polarity, formation and protrusion of lamellipodia, and the cell movements [40]. In contrast, low or oscillatory shear stress prevents cell migration. Via the activation of the anti-mitotic, AMP-activated protein kinase (AMPK) and of the proliferative Akt, high, unidirectional shear stress favors cell proliferation. 

## 5. Assessment of Endothelial Function in the Secondary Prevention

In a systematic review and meta-analysis, Matsuzawa et al. included a total of forty-one studies investigating the impact of conduit artery and microvascular endothelial function in the prognostic stratification of patients with a history of cardiovascular events [89]. The authors confirmed that peripheral endothelial function is a predictor of future cardiovascular events, across different subgroups, with a doubling in the risk by each 1 SD worsening, in each variable.

These data also allow hypothesizing that all mediators of endothelial function, and not exclusively NO, are involved in the vascular homeostasis. Indeed, although NO bioavailability mediated circa 50% of the flow-mediated dilation, it has a minor role in the microcirculation. Notably, the assessment of endothelial function has a predictive impact in different clinical settings—in coronary artery disease, an association between an impairment in endothelial function at 2–5 days, after the onset of an acute coronary syndrome, predicted subsequent CV events [90]. In the setting of heart failure, Shechter et al. provided evidence supporting the existence of a prognostic value of flow-mediated dilation in patients with New York Heart Association Class IV ischemic heart failure [91]. Finally, in the setting of stable coronary artery disease, peripheral endothelial dysfunction is associated with the risk of cardiovascular events, independent of the traditional risk factors and the coronary plaque complexity, as assessed by the SYNTAX Score [89]. Finally, the changes in the endothelial observed in response to different therapies, might provide important information to detect the effectiveness of the medications and develop individualized strategies. For instance, therapy with statins, b-blockers, ACE inhibitors, and several other drugs has been associated with improvements in the endothelial function (reviewed in [92]). Whether a therapeutic strategy, based on the assessment of endothelial functions, might result in an improved prognosis, needs to be tested in randomized trials. 

## 6. Conclusions

The monitoring of endothelial function has the theoretical potential to provide information on vascular health, including the impact of risk factors and therapies, at the different stages of the disease. In particular, these methods might serve to monitor the efficacy of medical therapies and evaluate responses at the individual patient level. Despite this sound rationale, however, practical considerations, including the limited reproducibility and high inter-subject variability of the methods, complicate their introduction in clinical routine.

## Figures and Tables

**Figure 1 ijms-19-03838-f001:**
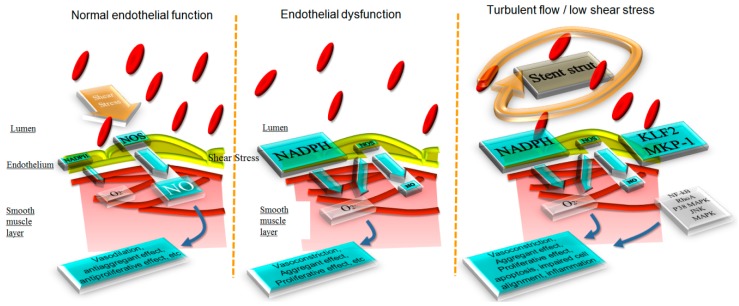
Schematic representation of the effect of shear stress on the endothelial production of nitric oxide and reactive oxygen species. In a normal endothelium, laminar shear stress stimulated nitric oxide production, with vasodilator, anti-inflammatory, and anti-proliferative effects. In the setting of endothelial dysfunction, among others, the production of reactive oxygen species inhibits these mechanisms. After stenting, the disruption of laminar shear stress, the mechanical trauma and the increased wall strain activate a series of mechanisms that simulate (and add to) endothelial dysfunction.

**Figure 2 ijms-19-03838-f002:**
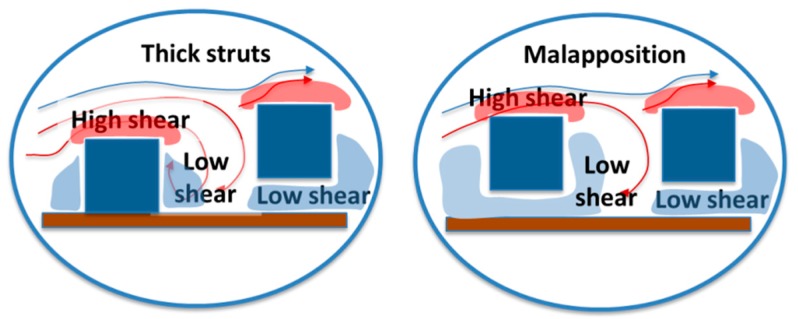
The shear stress modifications induced by stenting.

**Figure 3 ijms-19-03838-f003:**
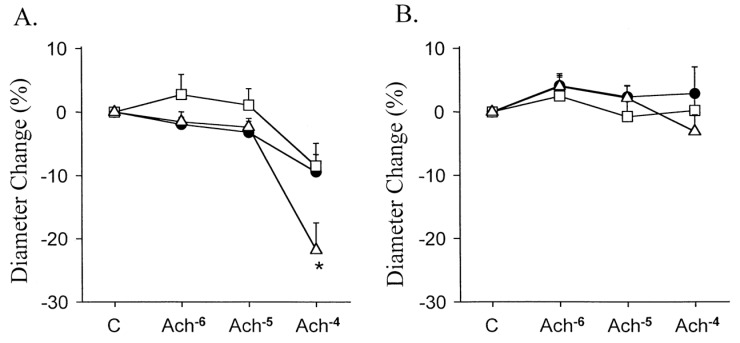
Response to the intracoronary infusions of acetylcholine, expressed as percentage change in mean luminal diameter of the left anterior descending coronary artery (**A**) and circumflex coronary artery (**B**). * *p* = 0.02 versus balloon angioplasty and directional atherectomy groups. Open triangle = Stent group. Closed circle = Balloon angioplasty group. Open square = Directional atherectomy group. With permission from [44].

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
