# Peer review of "Endothelial Function: A Short Guide for the Interventional Cardiologist"

_ijms, 2018, doi:10.3390/ijms19123838_

Round 1
Reviewer 1 Report
The review presented by Tommaso Gori is very well written, in which the endothelial function in coronary atherosclerosis is clearly concluded. The only suggestion for the manuscript is the figure1. The figure could be more informative with a more detailed description either from the legend or from the manuscript.
Author Response
Thank you very much for your comment. Figure 1 has been modified, and a better description has been added in the figure legend.
Reviewer 2 Report
The review by Gori, provides information on the endothelial dys/function in atherosclerotic plaque development and on the impact of stenting on endothelial function.
It is well written but few of the concepts need to be explained in detail when it comes to endothelial cells during cardiovascular diseases especially after coronary artery stenting.
1) There needs to be the inclusion of more detailed molecular mechanistic insights of endothelial dysfuntion during atherosclerotic plaque progression both in the introduction and in the first part of the review, ie., in ''Endothelial dysfunction as a mechanism and diagnostic/prognostic marker if coronary atherosclerosis''.
2) In the second part, while explaining the impact of coronary artery stenting on endothelial dysfunction, concepts are a bit superficial and a bit widely known. Author needs to give more information on the two different mechanisms the can explain endothelial dysfunction by DES, like, how biological effects of the drug as reviewer thinks that the information provided is too less. Please consider improving it. Even for hypersensitivity reactions by DES are need to be better explained.
Please add author's future perspectives on the topicc.
There are many spelling errors and please correct.
In abstract, avoid using ''in turn''...''individual patients respond' to individual patient responds'
In line 29, remove ''in turn'' and you may use other wording.
in line 28, ''presence of dysfunction'' to dyfunction.
Figure one needs to be changed as it is not clear, visually.
Line 104, edit ''who knows''.
In line 137, ''they function''
In line 257, regeneration and regrowth seem to be same
In line 304, ''potential to provide''.
Author Response
Thank you very much for your review.
1) There needs to be the inclusion of more detailed molecular mechanistic insights of endothelial dysfuntion during atherosclerotic plaque progression both in the introduction and in the first part of the review, ie., in ''Endothelial dysfunction as a mechanism and diagnostic/prognostic marker if coronary atherosclerosis''.
Thank you very much for this comment. We have expanded the section to include some more mechanisms, but we would also like to note that this is not the major focus of the paper, which we tried to focus on the impact of stenting, with an eye to the implications for the intervnetional cardiologist. We hope you will find this version acceptable.
2) In the second part, while explaining the impact of coronary artery stenting on endothelial dysfunction, concepts are a bit superficial and a bit widely known. Author needs to give more information on the two different mechanisms the can explain endothelial dysfunction by DES, like, how biological effects of the drug as reviewer thinks that the information provided is too less. Please consider improving it. Even for hypersensitivity reactions by DES are need to be better explained.
We have expanded this section to include the role of shear stress - and associated molecular mechanisms, as well as the role of strain. An additional figure has also been added.
> Please add author's future perspectives on the topicc.
this has been done.
> There are many spelling errors and please correct.
This has been done
> In abstract, avoid using ''in turn''...''individual patients respond' to individual patient responds'
This has been corrected
> In line 29, remove ''in turn'' and you may use other wording.
done
> in line 28, ''presence of dysfunction'' to dyfunction.
done
>Figure one needs to be changed as it is not clear, visually.
The figure has been modified to include endothelial function, dysfunction, and stenting
>Line 104, edit ''who knows''.
This has not been found, sorry
> In line 137, ''they function''
corrected
>In line 257, regeneration and regrowth seem to be same
corrected
>In line 304, ''potential to provide''.

Round 2
Reviewer 2 Report
It is much improved.
Minor
1) Please correct reference number arrangements carefully. Thats all.
2) Few wordings need to be corrected. Please re-read and correct.